# Wearable Devices and Smartphone Inertial Sensors for Static Balance Assessment: A Concurrent Validity Study in Young Adult Population

**DOI:** 10.3390/jpm12071019

**Published:** 2022-06-21

**Authors:** Luciana Abrantes Rodrigues, Enzo Gabriel Rocha Santos, Patrícia Seixas Alves Santos, Yuzo Igarashi, Luana Karine Resende Oliveira, Gustavo Henrique Lima Pinto, Bruno Lopes Santos Lobato, André Santos Cabral, Anderson Belgamo, Anselmo Athayde Costa e Silva, Bianca Callegari, Givago Silva Souza

**Affiliations:** 1Núcleo de Medicina Tropical, Universidade Federal do Pará, Belém 66050-540, Brazil; lucianaabrantesr@gmail.com (L.A.R.); yuzoigarashi@yahoo.com.br (Y.I.); 2Instituto de Ciências Exatas e Naturais, Universidade Federal do Pará, Belém 66050-540, Brazil; enzogabrielrocha29@gmail.com (E.G.R.S.); gpinto@ufpa.br (G.H.L.P.); 3Instituto de Ciências Biológicas, Universidade Federal do Pará, Belém 66050-540, Brazil; paguseixas@gmail.com; 4Instituto de Ciências da Saúde, Universidade Federal do Pará, Belém 66050-540, Brazil; luana.resende.oliveira@ics.ufpa.br (L.K.R.O.); bruls4@ufpa.br (B.L.S.L.); anselmocs@ufpa.br (A.A.C.e.S.); callegari@ufpa.br (B.C.); 5Centro de Ciências Biológicas e da Saúde, Universidade do Estado do Pará, Belém 66087-670, Brazil; ascfisio@gmail.com; 6Departamento de Ciência da Computação, Instituto Federal de São Paulo, Piracicaba 08021-090, Brazil; anderson@ifsp.edu.br; 7Instituto de Ciências da Educação, Universidade Federal do Pará, Belém 66050-540, Brazil

**Keywords:** static balance control, smartphone, wearable, accelerometer, posturography, concurrent validity

## Abstract

Falls represent a public health issue around the world and prevention is an important part of the politics of many countries. The standard method of evaluating balance is posturography using a force platform, which has high financial costs. Other instruments, such as portable devices and smartphones, have been evaluated as low-cost alternatives to the screening of balance control. Although smartphones and wearables have different sizes, shapes, and weights, they have been systematically validated for static balance control tasks. Different studies have applied different experimental configurations to validate the inertial measurements obtained by these devices. We aim to evaluate the concurrent validity of a smartphone and a portable device for the evaluation of static balance control in the same group of participants. Twenty-six healthy and young subjects comprised the sample. The validity for static balance control evaluation of built-in accelerometers inside portable smartphone and wearable devices was tested considering force platform recordings as a gold standard for comparisons. A linear correlation (r) between the quantitative variables obtained from the inertial sensors and the force platform was used as an indicator of the concurrent validity. Reliability of the measures was calculated using Intraclass correlation in a subsample (n = 14). Smartphones had 11 out of 12 variables with significant moderate to very high correlation (r > 0.5, *p* < 0.05) with force platform variables in open eyes, closed eyes, and unipedal conditions, while wearable devices had 8 out of 12 variables with moderate to very high correlation (r > 0.5, *p* < 0.05) with force platform variables under the same task conditions. Significant reliabilities were found in closed eye conditions for smartphones and wearables. The smartphone and wearable devices had concurrent validity for the static balance evaluation and the smartphone had better validity results than the wearables for the static balance evaluation.

## 1. Introduction

Balance is the basic ability to perform any movement, as well as daily activities, maintaining the center of mass (COM) within the support basis without losing equilibrium [1]. It is commonly classified as static balance, in which there is postural maintenance with minimal sway, or dynamic balance, that occurs during the execution of a motor skill that influences body orientation [2,3].

Currently, the most accurate instrument for objective quantification of static balance control evaluation is the force platform, which records the center of pressure displacements on the anteroposterior and mediolateral axes. However, due to the high cost and difficulty in handling this tool, it is mainly restricted to well-trained evaluators working in research centers [4]. Thus, to overcome these limitations, more accessible and portable tools can be used in an outpatient setting to assess static balance, such as smartphones or wearable devices that include built-in inertial sensors [5].

The use of smartphones or wearable devices with built-in inertial sensors is widespread, and several applications are being developed for the evaluation of static balance [6]. Many of them have validated the inertial measurements compared to some gold-standard methods. Table 1 lists several investigations that validated applications for portable devices with built-in inertial sensors during balance evaluation.

Usually, the validity of the application is carried out in open and closed eyes conditions, with a standard placement of the device in the lumbar spine, specifically the L5 vertebra, because it is close to the body’s center of mass. However, there is significant variability in other parameters, such as a device with an inertial sensor (smartphones with different dimensions, iPads, iPods, wearable devices) [19,20,21], the duration of inertial recordings (ranging between 10–90 s) [9,20,23], the gold standard instrument used to validate the inertial sensor (other inertial sensors, video capture, force platform or scales) [7,13,16], the quantitative measures of static balance assessment (time domain or frequency domain), and population for validity (healthy young subjects, elderly population, athletes, patients with Parkinson’s disease, with unilateral vestibular failure). It is not clear how these differences influence the validity of the results. A standardized clinical guideline for stabilometry indicates the force platform as a gold standard instrument, 45 s of recording, open and closed eye conditions, and feet position 15 cm parallel apart [24].

It is important to note that both smartphones and wearables represent a set of several instruments with differences in their physical features (size, weight) and available software for data analysis. It is not clear whether these differences could lead to different outcomes in the static balance control assessment. No study has validated inertial recordings obtained using smartphones and wearables during static balance evaluation in the same subjects compared to data obtained from a force platform. Our hypothesis was that smartphones have a similar validity to wearable devices for evaluating static balance control.

In the present investigation, our objective was to evaluate the concurrent validity of a homemade app, named Momentum, for smartphones and lightweight commercial wearables for static balance evaluation in the same subjects. The current validation includes three postural conditions (bipedal support with open eyes, bipedal support with closed eyes, and unipedal support with open eyes), two inertial devices (wearable and smartphone) and a force platform, and four quantitative stabilometric parameters (area of the center of pressure displacement, total displacement of the center of pressure, anteroposterior and mediolateral RMS amplitudes).

## 2. Materials and Methods

Study design

We investigated the concurrent validity of a smartphone and a lightweight portable wearable device.

Ethical considerations

The procedures performed were approved by the Research Ethics Committee of the Tropical Medicine Center of the Federal University of Pará (Report #633,187). All participants were informed about the investigation procedures and signed a consent form to participate in the study.

Subjects

The study was carried out in the Laboratory of Human Motricity Studies of the Institute of Health Sciences of the Federal University of Pará with 26 participants (15 females, 11 males, mean age 31 ± 7 years). No participants reported complaints or clinical alterations in balance. 

Static balance evaluation

Three devices were implemented to assess static balance: a force platform, a smartphone device, and a commercial wearable sensor. All three trials were performed during the static balance task for 60 s. Participants remained standing on the force platform, barefoot, with feet parallel to 15 cm apart, head straight, and gaze fixed on a point marked on the wall 1 m away [25]. In carrying out the test, three attempts were made, each lasting one minute, with the attempts being performed with eyes open and bipedal support, eyes closed and bipedal support, and eyes open and unipedal support. There was a 30-s rest interval between attempts. The smartphone and the commercial wearable device were attached to the participant’s body using an elastic strap in the lower lumbar spine region (L3–L5 level). Figure 1 shows a scheme of the placement of the inertial devices during the static balance control assessment. The recording conditions were randomly chosen.

The static balance was evaluated using a force platform as a gold-standard method [25] and two devices containing inertial sensors (a smartphone and a wearable device) as methods to be validated. The force platform (BIOMEC400, EMG System do Brasil, Ltd.a., São Paulo, Brazil) contained sensors distributed over 50 cm^2^, and a notebook with the Biomec program (EMG System do Brasil, Ltd.a., SP, Brazil) received its readings with an acquisition rate of 100 Hz. A smartphone device (Samsung Galaxy A10, 149.9 mm × 70.4 mm × 7.8 mm, 157 g, Samsung, Seoul, South Korea) was also used. Samsung Galaxy A10 has emerged as one of the best-selling Android devices and its dimensions are in the range of user preferences of mobile sizes [26]. We used a custom-made application (Momentum app) programmed in Android Studio to record the inertial signals from the triaxial accelerometer (LMS6DSL model, amplitude resolution: 16 bits, range: ±4 g) present on the phone device with an acquisition rate of 50 Hz. The Momentum app has been validated for different protocols of movement evaluation [27,28] and it is available for internal testing in the Play Store on request to the authors of the present investigation. The wearable commercial device (Model MetamotionC, MbientLab, San Francisco, CA, USA) was a portable inertial sensor that works through small-sized equipment (25 mm diameter × 4 mm in case, 5.6 g), enabling positioning in different body regions and requiring smartphone applications for Android and iOS systems (MetaBase, MbientLab, San Francisco, CA, USA) to store the recordings collected by the device. The wearable acquisition rate of the triaxial accelerometer (Bosch BMI 160 model, amplitude resolution: 16 bits, range: ±8 g) was 100 Hz.

Data analysis

Data were processed using computational routines developed in the MATLAB language (version R2020a, Mathworks, Nettick, MA, USA). The temporal series from the force platform and the inertial devices (smartphones and wearable devices) underwent initial treatment of detrend with later off-line filtering using a Butterworth second-order filter between 0.1 and 20 Hz. After starting the recording on the three devices, the participant was asked to jump over the force platform, and the jump signal was used to trigger the recordings in the computational routines of analysis. Time series were quantified in the time domain through parameters commonly associated with deficits in the balance control [8,29,30,31,32] as following [33]:

(i) RMS amplitude of the stabilograms in the mediolateral and anteroposterior axes (Equation (1)). For the force platform, this parameter was quantified in millimeters, and for the smartphone and the wearable device, this parameter was quantified in gravity units.
(1)$$RMS_{amplitude}=\sqrt{\sum_{i=1}^{n}{\left(X_{i}\right)}^{2}}$$
where $X_{i}$ is the reading of the device ith moment of the recording and n is the total number of readings of the recording in the anteroposterior or mediolateral axes.

(ii) The total deviation of the statokinesiogram represents the length of the trajectory of the center of pressure over the base of the support (Equation (2)). For the force platform, this parameter was quantified in millimeters, and for the smartphone and the wearable device, this parameter was quantified in gravity units.
(2)$$RMS_{amplitude}=\sum \sqrt{{\left(X_{AP}\right)}^{2}+{\left(X_{ML}\right)}^{2}}$$
where $X_{AP}$ is the vector of readings of the device in the anteroposterior axes and $X_{ML}$ is the vector of readings of the device in the mediolateral axes.

(iii) Statokinesiogram deviation area, which is also an indicator of the two-dimensional deviation of the center of pressure over the base of support (Equations (3) and (4)). For the force platform, this parameter was quantified in square millimeters, and for the smartphone and the wearable device, this parameter was quantified in square gravity units.
(3)$$\left[vec,val\right]=eig\left(cov\left(X_{AP},X_{ML}\right)\right)$$
where eig is the MATLAB function used to calculate the eigenvalues (val) and eigenvectors (vec) of the covariance of the vectors of readings in the anteroposterior and mediolateral axes calculated using the MATLAB function cov.
(4)$$area=pi*prod\left(2.4478\times \sqrt{svd\left(val\right)}\right)$$
where area is the confidence ellipse area, pi is 3.1415, prod is a MATLAB function to calculate the product of the array elements, svd is a MATLAB function to proceed single value decomposition.

No additional toolbox of MATLAB was needed to proceed the algorithms.

Statistics

The validity of the smartphone and the wearable device was based on Pearson’s product-moment correlation (r) between the parameters quantified in their time series and the same parameters obtained from the force platform (gold-standard method). Strength correlation was interpreted using the Cohen and Holiday criterion [34]: very high correlation (coefficient: from 0.9 to 1.0 or from −0.9 to −1.0); high correlation (coefficient: from 0.7 to 0.89 or from −0.7 to −0.89); moderate correlation (coefficient: from 0.4 to 0.69 or from −0.4 to −0.69); low correlation (coefficient: from 0.2 to 0.39 or from −0.2 to −0.39); and very low correlation (coefficient: below 0.19 or below −0.19). Parameters with correlations equal to or better than moderate were considered valid. Correlations with a *p*-value lower than 0.05 were considered significant. In the significant correlations, we calculated the power analysis of the correlation. The test-retest reliability of the measurements from each device was quantified by two-way random-absolute agreement model of intraclass correlation (ICC) coefficient in a subsample of the present investigation (n = 14). Those correlations with *p*-value lower than 0.05 were considered as significant. SPSS software was used to perform the statistics.

## 3. Results

### 3.1. Validity Analysis

Figure 2, Figure 3 and Figure 4 represent the recordings of body sway in open eyes, closed eyes, and unipedal support conditions from the participants using the force platform (upper panels), smartphone (intermediate panels), and wearable device (lower panels), respectively. Individual recordings are in gray on each panel, and the grand-averaged recordings are in black. In a visual inspection, we can observe large oscillations and variability along the anteroposterior axes for all devices and experimental conditions, as we expected.

Table 2 shows the mean values (and standard deviations) of the quantitative parameters of the static balance evaluation using each device under the different experimental conditions. Table 3 shows the Pearson product-moment correlation coefficients for the linear correlation between the quantitative parameters obtained from the force platform and the corresponding parameter estimated from the inertial sensor of the smartphone and the wearable device. The statistical significance of the correlation was considered an indicator of the validity of the inertial device. Figure 5 shows the correlation plots between the force platform measurements and the inertial measurements in the several experimental conditions.

For the smartphone, it was observed that 11 of 12 quantitative parameters had a significant correlation with those of the force platform. Under open-eye conditions, there were three moderate and significant correlations and one high and significant correlation, while under closed-eye conditions, there were three moderate correlations and one low and non-significant correlation. In unipedal support with open eyes, there were two very high and significant correlations, one high and significant correlation, and one moderate and significant correlation.

For the wearable device, it was observed that 8 of 12 parameters reached a significant correlation with those of the force platform. Of the parameters with significant correlations, there were three moderate correlations in the open eye condition; there was one moderate correlation in the closed eye condition; and there was a very high correlation, two high correlations and one moderate correlation for unipedal support.

### 3.2. Reliability Analysis

Table 4 shows the ICC value for the different parameters, devices, and experimental conditions. We observed that most of the parameters had non-significant reliability, except by total displacement (ICC = 0.81) and RMSX (ICC = 0.91) obtained using a smartphone in closed eyes condition, and total displacement (ICC = 0.03) obtained using a wearable device in closed eyes condition.

## 4. Discussion

The present study investigated the validity of inertial sensors present in smartphones and wearable devices to evaluate static balance control by comparing a gold standard method, force plate-based posturography. We used four variables (TD, AREA, RMSX, and RMSY) in 3 static balance control conditions (eyes open, eyes closed, and unipedal support) to test the validity of the inertial sensors. In general, our main findings confirmed inertial sensors as valid tools for evaluating static balance control as an alternative for force platform measurement in a young adult population. The present investigation also added that smartphones had better validity outcomes than lightweight wearables for static balance control assessment and that the greater the disturbances during the balance task, the better the validity of the instrument.

Several investigations have evaluated the validity of inertial sensors for static balance control, as shown in Table 1. Usually, investigations included some balance control conditions and many variables to validate inertial sensor measurements. Differences between investigations are found in the number of task conditions and inertial variables used for validity, the statistical method used to validate, and the type of inertial sensor used. Most investigations have used open and closed eyes during the static balance control evaluation, and few other conditions have been validated, such as the unipedal support used in the present study [6]. Many variables have been used to validate inertial measurements for the evaluation of static balance control, such as variables in the time domain (RMS, range, area, total displacement) and variables in the time frequency (median frequency, total power) variables [8,10,17,22]. As there are no systematic findings indicating a higher preference for any variable, we chose only time-domain variables for our analysis. The T test, Pearson’s product-moment correlation, and Spearman correlation were used to test the validity, and we followed Pearson’s product-moment correlation to be used as a criterion for the validity of the instruments.

To the best of our knowledge, this is the first description to validate multiple inertial sensor devices in the same group of participants. All participants performed the same task and were evaluated by the same experimenter, and body displacement was recorded simultaneously by all instruments. The reason for the differences in validity obtained from each inertial instrument is not clear. One potential factor that could explain these differences is the different masses and sizes of the instruments. Wearable devices are lightweight instruments and may be more susceptible to responding to different forces, which adds more noise to their recordings. Smartphones have dimensions of several centimeters, and their size allows a broader zone of attachment to the body. In our experiments, we searched for the ability to fix both instruments to the body as tightly as possible, avoiding discomfort to the participant. Previously, weak to moderate correlations were also found between inertial sensor-built wearable devices and force platform recordings [35]. It is confirmatory that we have obtained a moderate to high correlation between posturographic and inertial measurements for static balance control because many other studies have reported similar findings [36]. The highest correlations between inertial and force platform measurements were found in the unipedal support condition, which is expected to cause greater body displacement. We found moderate correlations between the open and closed-eye conditions because of the smaller amplitude of the body oscillations under these conditions. The validity of inertial measurement units was investigated for clinical movement analysis and smaller validity indexes were found in movements with a small range of motion [37]. It is important to highlight that each sensor has its own level of self-noise, offset, sensitivity, and bandwidth that also may introduce differences between sensors.

The limitation of the present investigation is low sample size, transferability of the current findings to other populations including the elderly and/or patients. Future investigations can extend the number of subjects in validity and reliability quantification, as well as apply to other populations and clinical conditions.

We conclude that smartphones and lightweight wearable devices had validity to evaluate static balance control under different experimental conditions. The validity of smartphones to evaluate static balance control candidates confirms them as a low-cost alternative to the force platform and encourages their use in routine clinical practice, in health services for low-income populations, and for screening purposes. This makes it possible to popularize a quantitative assessment of static balance control that can be fundamental in identifying functional losses in different populations.

## Figures and Tables

**Figure 1 jpm-12-01019-f001:**
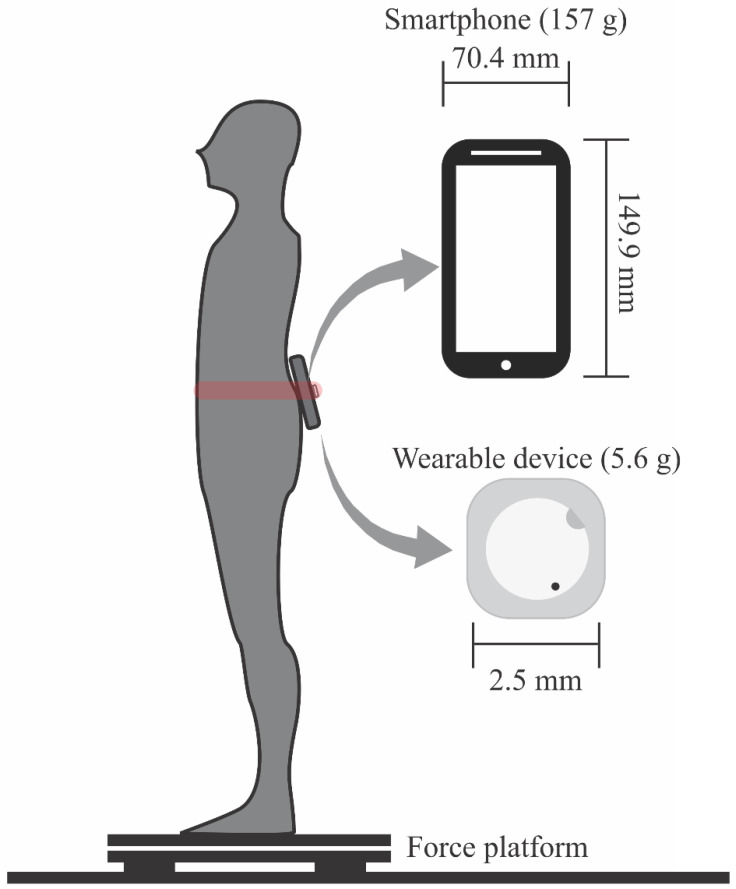
Scheme of the experimental setup. The participant was requested to keep in a quite standing position on a force platform. A smartphone and wearable were fixed in the lumbar region using an elastic strap. Panels highlight the dimensions of the smartphone and wearable device.

**Figure 2 jpm-12-01019-f002:**
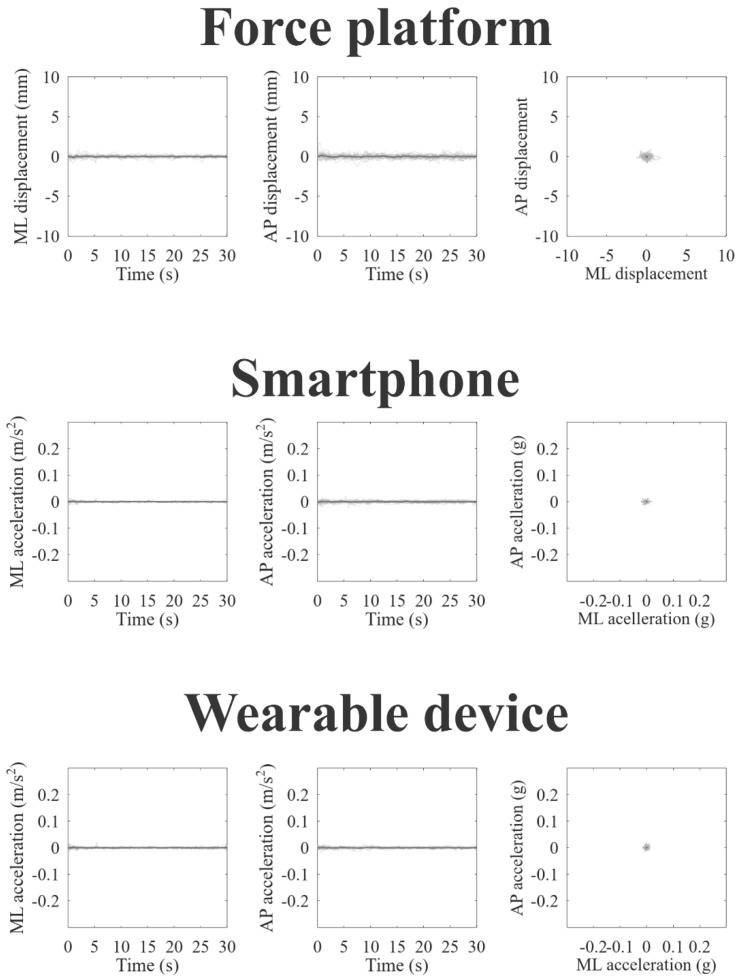
Static balance assessment in open eyes condition obtained from the force platform (upper line plots), smartphone (intermediate line plots), and wearable device (lower line plots). For all instruments is shown stabilogram on the mediolateral axes (in the left); stabilogram on the anteroposterior axes; (in the center), and statokinesiogram (in the right) The black lines represent the grand mean recording, while the gray lines represent the individual recordings.

**Figure 3 jpm-12-01019-f003:**
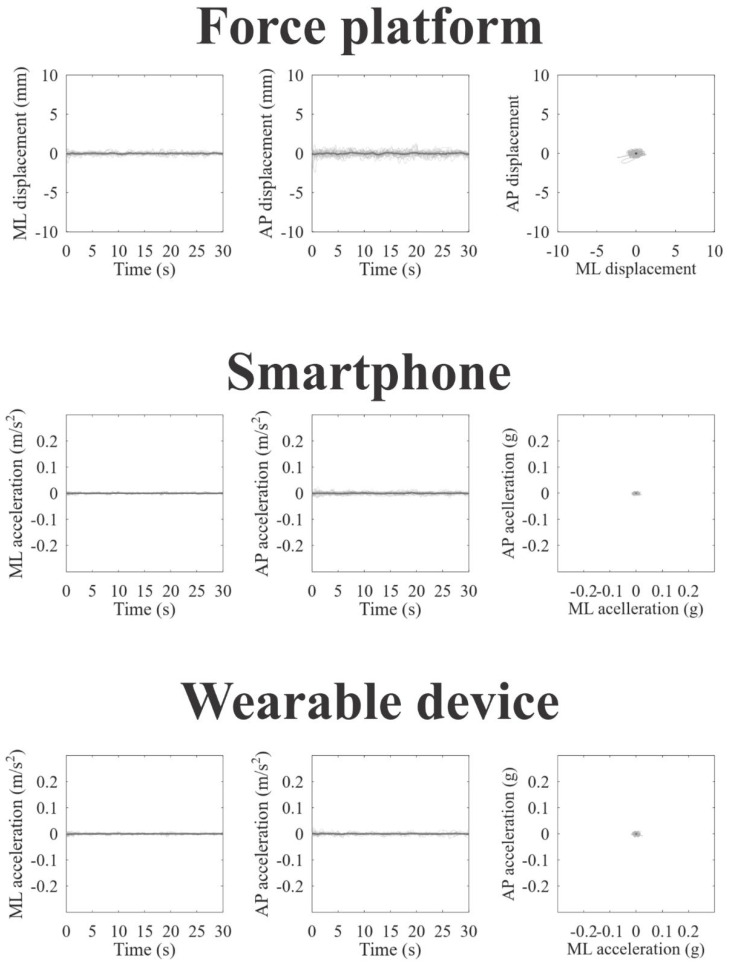
Static balance assessment in closed eyes condition obtained from the force platform (upper line plots), smartphone (intermediate line plots), and wearable device (lower line plots). For all instruments is shown stabilogram on the mediolateral axes (in the left); stabilogram on the anteroposterior axes; (in the center), and statokinesiogram (in the right) The black lines represent the grand mean recording, while the gray lines represent the individual recordings.

**Figure 4 jpm-12-01019-f004:**
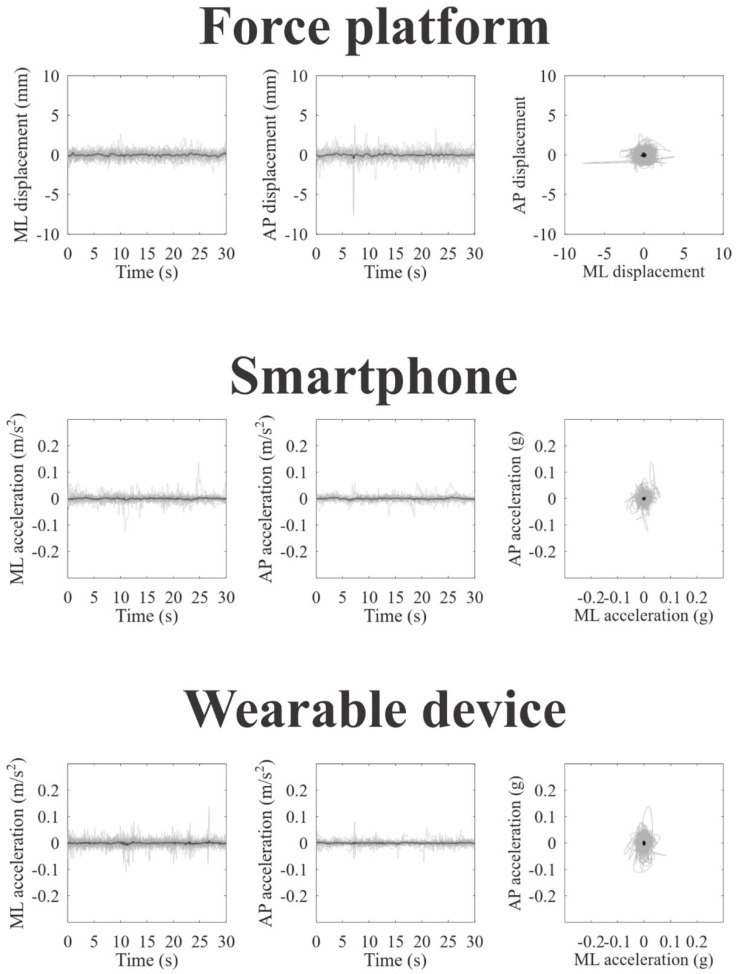
Static balance assessment in unipedal support condition obtained from the force platform (upper line plots), smartphone (intermediate line plots), and wearable device (lower line plots). For all instruments is shown stabilogram on the mediolateral axes (in the left); stabilogram on the anteroposterior axes; (in the center), and statokinesiogram (in the right) The black lines represent the grand mean recording, while the gray lines represent the individual recordings.

**Figure 5 jpm-12-01019-f005:**
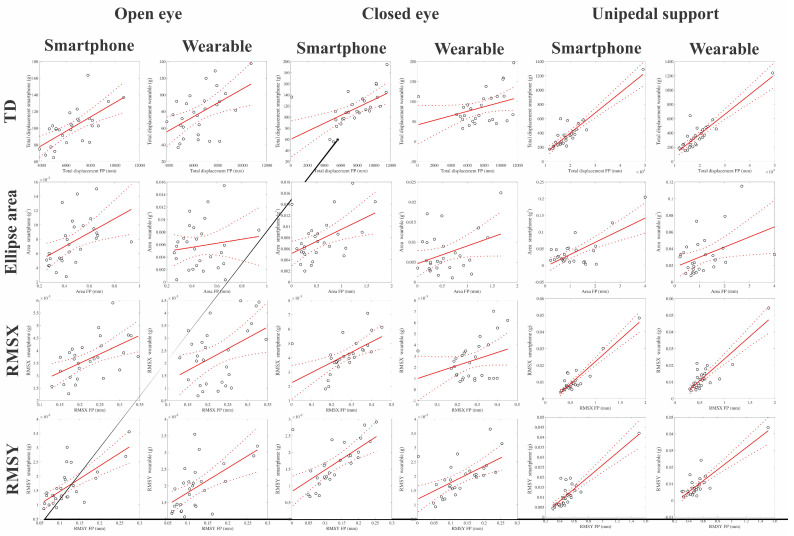
Correlation plots between the measurements obtained by force platform (x-axes) and inertial devices (y-axes). Continuous red lines represent the linear correlation and dashed red lines represent the confidence interval of the correlation.

**Table 1 jpm-12-01019-t001:** List of references that validated inertial sensors for balance assessment.

Reference	Inertial Sensor (AR Hz)	Recording Duration	Gold-Standard Instrument
Whitney et al. (2011) [7]	Wearable (100 Hz)	90 s	Force platform
Mancini et al. (2012) [8]	Wearable (50 Hz)	30 min	Force Platform
Seimetz et al. (2012) [9]	Wearable (n.i.)	90 s	Force platform
Ozinga et al. (2014) [10]	iPad 3 (100 Hz)	60 s	Motion capture
Patterson et al. (2014) [11]	iPod touch (60 Hz)	20 s	Balance Error Scoring System
Rouis et al. (2014) [12]	Wearable (50 Hz)	30 s	Force platform
Alberts et al. (2015) [13]	iPad (100 Hz)	20 s	Motion capture
Abe et al. (2015) [14]	Wearable (50 Hz)	30 s	Motion capture
Heebner et al. (2015) [15]	Wearable (1000 Hz)	30 min	Force platform
Kosse et al. (2015) [16]	iPod touch (88–92 Hz)	60 s	Wearable
Neville et al. (2015) [17]	Wearable (250 Hz)	30 s	Motion capture, force platform
Alessandrini et al. (2017) [18]	Wearable (25 Hz)	60 s	Force Platform
Burghart et al. (2017) [19]	iPod Touch (10 Hz)	60 s	Force platform
Dabbs et al. (2017) [20]	Iphone (n.i.)	10 s	Force platform
Fiems et al. (2018) [21]	iPod Touch (60 Hz)	1 h	Fall protocol and Modified Clinical, Test of Sensory Integration and Balance Protocol
Kim et al. (2018) [22]	Wearable (50 Hz)	30 s	Motion capture
Hsieh et al. (2019) [23]	Smartphone (200 Hz)	30 s	Force Platform

AR: acquisition rate. n.i.: not informed.

**Table 2 jpm-12-01019-t002:** Mean values of the stabilometric parameters estimated using the force platform and accelerometers in wearable and mobile devices.

Parameter	Force Platform		Smartphone		MetaMotionC^®^	
	Mean	SD	Mean	SD	Mean	SD
Open eyes						
TD	6443.88	1701.04	98.12	18.6	74.54	24.53
AREA	0.47	0.16	0.0076	0.0032	0.0054	0.0036
RMSX	0.22	0.06	0.0036	0.0006	0.0022	0.0011
RMSY	0.12	0.05	0.0016	0.0006	0.0019	0.0006
Closed eyes						
TD	7324.83	2420.43	109.41	26.39	77.49	32.21
AREA	0.54	0.37	0.0073	0.0037	0.0061	0.0038
RMSX	0.26	0.09	0.004	0.001	0.0023	0.0014
RMSY	0.12	0.06	0.0016	0.0006	0.0019	0.0006
Unipedal support						
TD	19575.9	7073.5	383.08	222.02	365.69	221.91
AREA	1.17	0.8519	0.0384	0.0444	0.0317	0.0246
RMSX	0.59	0.316	0.0111	0.0093	0.0131	0.01
RMSY	0.52	0.2239	0.0118	0.0077	0.0091	0.0086

TD:. Total deviation; AREA:. Statokinesiogram displacement area; RMSX: RMS amplitude in the mediolateral axes; RMSY: RMS amplitude on the anteroposterior axes. SD: standard deviation.

**Table 3 jpm-12-01019-t003:** Pearson’s correlation results between the force platform and the inertial sensors of smartphones and wearable devices. In the significant correlations, the two-sided power analysis is shown inside the parentheses.

Correlation	Smartphone	*p*-Value	MetaMotionC	*p*-Value
Open eyes				
TD	0.65 (0.99)	0.0001 *	0.51 (0.75)	0.0087 *
AREA	0.49 (0.75)	0.0105 *	0.14	0.5631
RMSX	0.54 (0.82)	0.0041 *	0.44 (0.62)	0.0238 *
RMSY	0.72 (0.99)	0.0001 *	0.58 (0.88)	0.0024 *
Closed eyes				
TD	0.57 (0.68)	0.0161 *	0.35	0.478
AREA	0.49	0.0565	0.36	0.3234
RMSX	0.57 (0.7)	0.0124 *	0.32	0.5053
RMSY	0.63 (0.9)	0.0015 *	0.58 (0.82)	0.0045 *
Unipedal support				
TD	0.91 (0.99)	0.0001 *	0.90 (0.99)	0.0001 *
AREA	0.70 (0.99)	0.0001 *	0.42 (0.57)	0.0339 *
RMSX	0.93 (0.99)	0.0001 *	0.87 (0.99)	0.0001 *
RMSY	0.87 (0.99)	0.0001 *	0.86 (0.99)	0.0001 *

TD:. Total deviation; AREA:. Statokinesiogram displacement area; RMSX: RMS amplitude in the mediolateral axes; RMSY:. RMS amplitude on the anteroposterior axes. * *p* < 0.05.

**Table 4 jpm-12-01019-t004:** Test- retest reliability (*p*-value) of the parameters obtained by force platform, smartphone, and wearable in the different experimental conditions in a subsample of the present investigation (n = 14).

Parameter	Open Eyes	Closed Eyes	Unipedal Support
Force platform			
TD	0.46 (0.14)	−0.15 (0.6)	0.54 (0.06)
AREA	0.54 (0.08)	0.12 (0.38)	0.32 (0.26)
RMSX	0.59 (0.07)	0.16 (0.37)	0.31 (0.23)
RMSY	0.2 (0.34)	0.2 (0.36)	0.47 (0.08)
Smartphone			
TD	0.22 (0.34)	0.81 (0.003)	0.3 (0.25)
AREA	−0.15 (0.6)	0.32 (0.22)	−0.01 (0.51)
RMSX	0.36 (0.21)	0.91 (0.001)	0.1 (0.42)
RMSY	0.18 (0.37)	0.47 (0.14)	0.34 (0.21)
Wearable device			
TD	−0.1 (0.59)	0.47 (0.03)	0.43 (0.17)
AREA	−0.23 (0.64)	0.13 (0.4)	0.17 (0.38)
RMSX	0.005 (0.49)	0.2 (0.2)	0.18 (0.32)
RMSY	0.59 (0.07)	0.58 (0.07)	0.25 (0.32)

TD:. Total deviation; AREA:. Statokinesiogram displacement area; RMSX: RMS amplitude on the mediolateral axes; RMSY:. RMS amplitude on the anteroposterior axes.

## Data Availability

The datasets used and/or analyzed during the current study are available from the corresponding author on reasonable request.

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
