# Peer review of "Wearable Devices and Smartphone Inertial Sensors for Static Balance Assessment: A Concurrent Validity Study in Young Adult Population"

_jpm, 2022, doi:10.3390/jpm12071019_

Round 1

Reviewer 1 Report

The study aims to assess validity of consumer-grade smartphone and research-grade wearable device for measurement of static balance. The device recordings are compared again readings from the force platform. The manuscript is well written while the results are easy to interpret. Please see my other remarks below.

Description of wearable device and smartphone: For completeness of description, please provide measurement range and amplitude resolution of accelerometer sensors.

The description (Lines 153-156) of Equation 4 is difficult to understand. Please paraphrase or break it down into small chunks. Also, please note if any of the MATLAB functions required installation of additional toolboxes.

Line 166: “below -0.19” did you mean “above -0.19”?

Line 168: ““Biostat software” What specific software did you use?

Figures 1-3: To ease comparison between scenarios (Figures), please resize x- and y-axes to the same range.

Figure 3: Please verify whether sways in ML displacement of the force platform in fact correspond to sways in AP acceleration in smartphone and wearable device.

Lines 254+: Please note that each sensor has its own level of self-noise, offset, sensitivity, and bandwidth. These factors might introduce differences between sensors in your study.

Author Response

Reviewer #1

The study aims to assess validity of consumer-grade smartphone and research-grade wearable device for measurement of static balance. The device recordings are compared again readings from the force platform. The manuscript is well written while the results are easy to interpret. Please see my other remarks below. 

  1. Description of wearable device and smartphone: For completeness of description, please provide measurement range and amplitude resolution of accelerometer sensors.
  2. Done. We added to the Methods section the information about model of the accelerometer, amplitude resolution, amplitude range, and rate of acquisition (Line 128-141).
  1. The description (Lines 153-156) of Equation 4 is difficult to understand. Please paraphrase or break it down into small chunks. Also, please note if any of the MATLAB functions required installation of additional toolboxes.
  2. Done.
  1. Line 166: “below -0.19” did you mean “above -0.19”?
  2. Yes, I did. Thanks. Done.
  1. Line 168: ““Biostat software” What specific software did you use?
  2. Thanks for the comment. We used SPSS software and it is informed in the revised version (Line 195).
  1. Figures 1-3: To ease comparison between scenarios (Figures), please resize x- and y-axes to the same range.
  2. Done.
  1. Figure 3: Please verify whether sways in ML displacement of the force platform in fact correspond to sways in AP acceleration in smartphone and wearable device.
  2. Thanks for the comment. We corrected the figures of the statokinesiograms and replaced in the Results section.
  1. Lines 254+: Please note that each sensor has its own level of self-noise, offset, sensitivity, and bandwidth. These factors might introduce differences between sensors in your study.
  2. Thanks for the comment. We added this comment in the Discussion section (Line 309).

Reviewer 2 Report

In this study, the authors have evaluated the feasibility of using portable devices and smartphones, as low-cost alternatives to gold the standard method, force plate-based posturography, for screening of balance control. The objective of the study is relevant but I have several comments below:

Title: I would suggest including ‘on young population’ in the title.

Introduction:

The authors mentioned that ‘However, there is significant variability in other parameters, such as the duration of inertial recording, the gold standard instrument used to validate the inertial sensor, and the quantitative measures of static balance assessment’. It might make sense to slightly expand this section by comparing the previous pieces of literature, their findings, and their limitations.

Method:

Under the ‘Subjects’ section: It would be important to mention your inclusion and exclusion criteria for recruiting participants. Also, you might explain, either statistically (for example by Power Analysis), or according to the previous literature, why you come up with 26 participants.

Under the ‘Static balance evaluation’ section: please provide a figure with the details of the used/developed setup. It is also important to explain why you’ve selected this specific smartphone device (Samsung Galaxy A10). It would furthermore be good to talk about the (Momentum app); Is it a custom-made app and will be possibly freely available?. Moreover, it would be good to provide a citation for the ‘force platform as a gold-standard method’.

Under the ‘Data analysis’ section: please explain why you used RMS amplitude, the total deviation, and Statokinesiogram deviation area, as the outcome? In the discussion section, you have referred to other methods/parameters and there should be a logic behind and should be explained.

Under the ‘Statistics’ section: for the validity of a technique, it is very important to include also test-retest reliability analysis. I would highly suggest including this analysis in the current version of the manuscript, even if you have no access to all recruited participants (50-60% would be enough).

Results:

It is important, in addition to the current figure, to include also correlation figures (scatter plot), some main/significant correlations can be in the main text and the rest in the supplementary materials.

Discussion:

Please clearly mention in the first para of the discussion that the analysis and findings were done on healthy young populations.

There should be a para explaining the limitations and future directions of the study. For example, the low sample size, transferability of the current findings to other populations including the elderly and/or patients, and lack of test-retest reliability.

Conclusion: please add a conclusion section at the end of the study.

Author Response

In this study, the authors have evaluated the feasibility of using portable devices and smartphones, as low-cost alternatives to gold the standard method, force plate-based posturography, for screening of balance control. The objective of the study is relevant but I have several comments below:

  1. Title: I would suggest including ‘on young population’ in the title.
  2. Done.

Introduction:

  1. The authors mentioned that ‘However, there is significant variability in other parameters, such as the duration of inertial recording, the gold standard instrument used to validate the inertial sensor, and the quantitative measures of static balance assessment’. It might make sense to slightly expand this section by comparing the previous pieces of literature, their findings, and their limitations.
  2. Done (Lines 63-75).

Method:

  1. Under the ‘Subjects’ section: It would be important to mention your inclusion and exclusion criteria for recruiting participants. Also, you might explain, either statistically (for example by Power Analysis), or according to the previous literature, why you come up with 26 participants.
  2. Done. We added the power analysis of the linear correlation for each significant correlation.
  1. Under the ‘Static balance evaluation’ section: please provide a figure with the details of the used/developed setup.
  2. Done. We added a new Figure 1 with the experimental setup we used.
  1. It is also important to explain why you’ve selected this specific smartphone device (Samsung Galaxy A10).
  2. Done. We added the explanation in the Methods section (Line 129-131).
  1. It would furthermore be good to talk about the (Momentum app); Is it a custom-made app and will be possibly freely available?
  2. Done. We added the explanation in the Methods section (Lines 133-136).
  1. Moreover, it would be good to provide a citation for the ‘force platform as a gold-standard method’.
  2. Done (Line 124).
  1. Under the ‘Data analysis’ section: please explain why you used RMS amplitude, the total deviation, and Statokinesiogram deviation area, as the outcome? In the discussion section, you have referred to other methods/parameters and there should be a logic behind and should be explained.
  2. Done. We added to the Methods section that time series were quantified in the time domain through parameters commonly associated to deficits in the balance control (Line 151).
  1. Under the ‘Statistics’ section: for the validity of a technique, it is very important to include also test-retest reliability analysis. I would highly suggest including this analysis in the current version of the manuscript, even if you have no access to all recruited participants (50-60% would be enough).
  2. Done.

Results:

  1. It is important, in addition to the current figure, to include also correlation figures (scatter plot), some main/significant correlations can be in the main text and the rest in the supplementary materials.
  2. Done.

Discussion:

  1. Please clearly mention in the first para of the discussion that the analysis and findings were done on healthy young populations.
  2. Done.
  1. There should be a para explaining the limitations and future directions of the study. For example, the low sample size, transferability of the current findings to other populations including the elderly and/or patients, and lack of test-retest reliability.
  2. Done (Line 312-315).
  1. Conclusion: please add a conclusion section at the end of the study.
  2. Done.